# DISENTANGLING RECURRENT NEURAL DYNAMICS WITH STOCHASTIC REPRESENTATIONAL GEOMETRY

**David Lipshutz**[*][1]**, Amin Nejatbakhsh**[*][1]**, Alex H Willliams**[1,2]
[1]Center for Computational Neuroscience, Flatiron Institute
[2]Center for Neural Science, New York University
`{dlipshutz,anejatbakhsh,awilliams}@flatironinstitute.org`

## ABSTRACT

Uncovering and comparing the dynamical mechanisms that support neural processing remains a key challenge in the analysis of biological and artificial neural systems. However, measures of representational (dis)similarity in neural systems often assume that neural responses are static in time. Here, we show that stochastic shape distances (SSDs; Duong et al., 2023), which were developed to compare noisy neural responses to static inputs and lack an explicit notion of temporal structure, are well equipped to compare noisy dynamics. In two examples, we use SSDs, which interpolate between comparing mean trajectories and second-order fluctuations about mean trajectories, to disentangle recurrent versus external contributions to noisy dynamics.

## 1 INTRODUCTION

Biological and artificial neural networks represent external stimuli and actions in high-dimensional feature spaces. While different neural systems utilize distinct feature spaces, the geometry of representations in these spaces can be similar across systems (Dwivedi & Roig, 2019; Chung & Abbott, 2021), hinting that underlying computational principles may be shared.

Numerous distance measures between neural representations have been utilized, including: Representational Similarity Analysis (RSA, Kriegeskorte et al., 2008), Centered Kernel Alignment (CKA, Kornblith et al., 2019), Procrustes shape distance (Williams et al., 2021), and individual matching of neurons (Li et al., 2015; Khosla & Williams, 2023). Nearly all of these methods consider neural responses that are static and non-stochastic. That is, neural systems are idealized as a deterministic functions $f : \mathcal{S} \mapsto \mathbb{R}^N$ that map from a stimulus space $\mathcal{S}$ to $N$-dimensional feature space (where $N$ is the number of neurons in the system).

But many biological and artificial neural systems are neither deterministic nor static. In many brain regions, the mean neural response is often smaller than variance across trials (Goris et al., 2014), and shared trial-to-trial fluctuations (so called "noise correlations") are thought to be a crucial determinant of a circuit's signalling capacity (Averbeck et al., 2006). Moreover, neural responses unfold dynamically over time, with rich interplay of feedforward and recurrent interactions (Vyas et al., 2020). Of course, many artificial network models also contain components that are recurrent (e.g. RNNs) or stochastic (e.g. variational autoencoders).

Ignoring the stochastic and dynamical aspects of neural circuits may be a justified simplification in certain settings, but it is easy to imagine situations where traditional representation (dis)similarity measures are blind to important details. These deficiencies were separately addressed in two recent works by Duong et al. (2023) and Ostrow et al. (2023). The former proposed stochastic shape distances (SSDs) to geometrically quantify differences in trial-to-trial noise across networks without addressing recurrent dynamics. The latter utilized Koopman operator theory to develop Dynamical Similarity Analysis (DSA), which aims to quantify similarity in dynamical elements of flow fields without an explicit focus on noise statistics.

Here, we investigate the extent to which these two principles—stochasticity and recurrent dynamics—are interrelated in the context of quantifying representational similarity. The relation

---

[*]Equal contribution

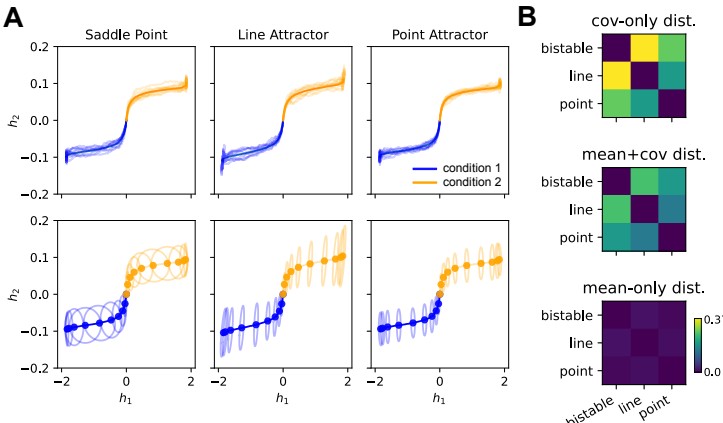

**Figure 1:** Demonstration that stochastic shape distances disambiguate recurrent flow fields. *(A)* Simulations of three systems with identical trial-average behavior but distinct recurrent dynamics. See main text and Appendix A.1 for details. *(B)* Pairwise SSD distances between the three systems in panel A, using eq. (1) with $\alpha = 0$ (top), $\alpha = 1$ (middle), and $\alpha = 2$ (bottom).

between the two has already been remarked upon in the broader literature. A recent paper by Galgali et al. (2023) demonstrates how stochastic deviations from an average dynamical trajectory ("residual dynamics") can be used to infer a dynamical flow field under certain conditions. It is also well understood that driving a deterministic system with noise (or some other "persistently exciting" signal) is necessary to infer its governing dynamics (Åström & Torsten, 1965; Mareels, 1984; Green & Moore, 1986). We show that SSDs, which lack any explicit notion of dynamics, can nonetheless be used to quantify differences arising from recurrent neural interactions.

## 2 RESULTS

We focus on SSDs that account for the mean and covariance of neural responses and ignore higher-order moments.[1] Specifically, consider network $A$ with mean responses $\{\boldsymbol{\mu}_1^{(A)}, \ldots, \boldsymbol{\mu}_M^{(A)}\}$ and covariances $\{\boldsymbol{\Sigma}_1^{(A)}, \ldots, \boldsymbol{\Sigma}_M^{(A)}\}$ in an $N$-dimensional feature space across $M$ conditions or "landmarks." The Wasserstein $\alpha$-SSD between $A$ and a similarly defined network $B$ is:

$$d_\alpha^2(A, B) = \min_{\boldsymbol{Q} \in \mathcal{O}(N)} \sum_{m=1}^{M} \left\{ \alpha \|\boldsymbol{\mu}_m^{(A)} - \boldsymbol{Q}\boldsymbol{\mu}_m^{(B)}\|_2^2 + (2-\alpha)\mathcal{B}^2(\boldsymbol{\Sigma}_m^{(A)}, \boldsymbol{Q}\boldsymbol{\Sigma}_m^{(B)}\boldsymbol{Q}^\mathsf{T}) \right\} \quad (1)$$

where $\mathcal{B}(\cdot, \cdot)$ is the *Bures distance* between positive semidefinite matrices, $\mathcal{O}(N)$ is the set of orthogonal matrices, and $0 \leq \alpha \leq 2$ determines the relative weight placed on differences between the means or covariances of the neural responses. In the original conception of SSDs, each landmark corresponds to a different network input—i.e., $m$ is an index over stimulus variables, such as natural images fed into a deep network. Below we will use $m$ to index over timebins or dynamic behavioral states. Equation (1) is described in more detail in section 2.4 of Duong et al. (2023).

### 2.1 DISTINGUISHING SYSTEMS WITH THE SAME TRIAL-AVERAGE TRAJECTORIES

We begin by showing that eq. (1) can be used "off-the-shelf" to distinguish between different recurrent dynamics that give rise to the same trial-average trajectories. To demonstrate, we ran simulations of three network dynamics following the construction by Galgali et al. (2023) and later leveraged by Ostrow et al. (2023). In these simulations, the recurrent dynamics are distinct–the first system is bistable, the second implements line attractor dynamics, and the third implements point attractor dynamics. However, the time varying input drive is tuned to render the trial-average responses identical. Figure 1A shows simulated dynamics under two conditions (yellow vs. blue),

---

[1]An alternative SSD based on energy distance (instead of Wasserstein distance) may be used to investigate differences in higher-order moments (see Duong et al., 2023). But we leave this possibility to future work.

corresponding to a binary decision outcome (see Galgali et al., 2023, for context). The top panels show single-trial dynamics (semi-transparent trajectories) and the trial-average. The bottom panels illustrate the trial-average (dots) and the marginal, across-trial covariance (ellipses) at individual time points. Full simulation details are provided in Appendix A.

It is clear from inspection that the three systems can be distinguished from their covariances, but not their trial averages. To confirm, we computed SSDs using eq. (1) treating timebins in distinct conditions as "landmarks." Concretely, we have $M = CT$ landmarks where $T$ is the number of timebins in a trial and $C = 2$ is the number of conditions (blue vs. yellow trajectories). The SSD with $\alpha = 2$ fails to distinguish the three cases, as expected (Figure 1B, mean-only distance). In contrast, the SSD with $\alpha = 0$ or $\alpha = 1$ successfully distinguishes the three systems (Figure 1B, mean + covariance and covariance-only distance). This reproduces one of the main capabilities of DSA (Fig. 3 in Ostrow et al., 2023). Indeed, we qualitatively reproduce their finding that the line and point attractor systems are most similar (i.e. lower distance relative to the bistable system).

## 2.2 WHY ACROSS-TRIAL COVARIANCE CAN DISTINGUISH RECURRENT DYNAMICS

Figure 1 shows a simple example where the shape of trial-to-trial "noise" can capture differences in recurrent dynamics, even when input-driven dynamics are tuned to make firing rate trajectories as similar as possible. In this section, we provide a theoretical explanation for this result which holds for any stochastic differential equation that takes the form:

$$d\boldsymbol{x}(t) = f(\boldsymbol{x}(t))dt + \boldsymbol{u}(t)dt + g(\boldsymbol{x}(t))d\boldsymbol{w}(t), \tag{2}$$

where $\boldsymbol{x}(t)$ is an $N$-dimensional vector of neural activities at time $t$, $f$ is a sufficiently regular (e.g., Lipschitz continuous) function governing time-invariant recurrent dynamics, $\boldsymbol{u} : \mathbb{R} \mapsto \mathbb{R}^N$ defines a deterministic time-varying input drive to the system, $g : \mathbb{R}^N \mapsto \mathbb{R}^{N \times N}$ is a sufficiently regular function that defines the noise structure, and $\boldsymbol{w}(t)$ is an $N$-dimensional standard Brownian motion. The stochastic process $\{\boldsymbol{x}(t)\}$ defines a family of distributions on $\mathbb{R}^N$ parameterized by $t$. Define the first two moments (mean and covariance) of these distributions as:

$$\boldsymbol{m}(t) := \mathbb{E}[\boldsymbol{x}(t)], \qquad \text{and} \qquad \boldsymbol{P}(t) := \mathbb{E}\left[(\boldsymbol{x}(t) - \boldsymbol{m}(t))(\boldsymbol{x}(t) - \boldsymbol{m}(t))^\mathsf{T}\right]. \tag{3}$$

We drop the dependence on $t$ below for brevity. Using standard methods from stochastic calculus (see e.g. Särkkä & Solin, 2019), one can show that $\boldsymbol{m}(t)$ and $\boldsymbol{P}(t)$ evolve according to the differential equations

$$\frac{d\boldsymbol{m}}{dt} = \mathbb{E}[f(\boldsymbol{x})] + \boldsymbol{u}, \qquad \frac{d\boldsymbol{P}}{dt} = \mathbb{E}[f(\boldsymbol{x})(\boldsymbol{x} - \boldsymbol{m})^\mathsf{T}] + \mathbb{E}[(\boldsymbol{x} - \boldsymbol{m})f(\boldsymbol{x})^\mathsf{T}] + \frac{1}{2}\mathbb{E}[g(\boldsymbol{x})g(\boldsymbol{x})^\mathsf{T}].$$

These differential equations provide a concrete explanation for the success of SSDs in Figure 1. Specifically, we see that the evolution of the mean trajectory $\boldsymbol{m}(t)$ depends on the input drive $\boldsymbol{u}(t)$, whereas the evolution of the covariance matrix $\boldsymbol{P}(t)$ *does not* explicitly depend on $\boldsymbol{u}(t)$. These equations suggest that SSDs (with $\alpha = 2$) are a useful metric for isolating differences in recurrent dynamics, while deterministic shape distances ($\alpha = 0$) are useful for comparing the joint effect of recurrent dynamics and input drive. We caution, however, that this does not guarantee that SSDs will always distinguish between networks with different nonlinear recurrent interactions.

To further elucidate the relationship between the covariance trajectories and the recurrent dynamics, it is instructive to consider the linear setting with additive noise—i.e., when $f(\boldsymbol{x}) = \boldsymbol{A}\boldsymbol{x}$ and $g(\boldsymbol{x}) = \sqrt{2}\boldsymbol{L}$ for $N \times N$ matrices $\boldsymbol{A}$ and $\boldsymbol{L}$. In this case, the evolution of the covariance $\boldsymbol{P}(t)$ is governed by the linear differential equation:

$$\frac{d\boldsymbol{P}}{dt} = \boldsymbol{A}\boldsymbol{P} + \boldsymbol{P}\boldsymbol{A}^\mathsf{T} + \boldsymbol{L}\boldsymbol{L}^\mathsf{T}.$$

In particular, it is clear that the evolution of the covariance matrix depends only on the recurrent dynamics matrix $\boldsymbol{A}$ and noise matrix $\boldsymbol{L}\boldsymbol{L}^\mathsf{T}$, not the time-varying input drive $\boldsymbol{u}(t)$.

## 2.3 DISTINGUISHING SYSTEMS WITHOUT TRIAL STRUCTURE

As neuroscientists study more complex and naturalistic behaviors, it has become increasingly common for experimental setups to have less trial to trial structure (Williams & Linderman, 2021). How

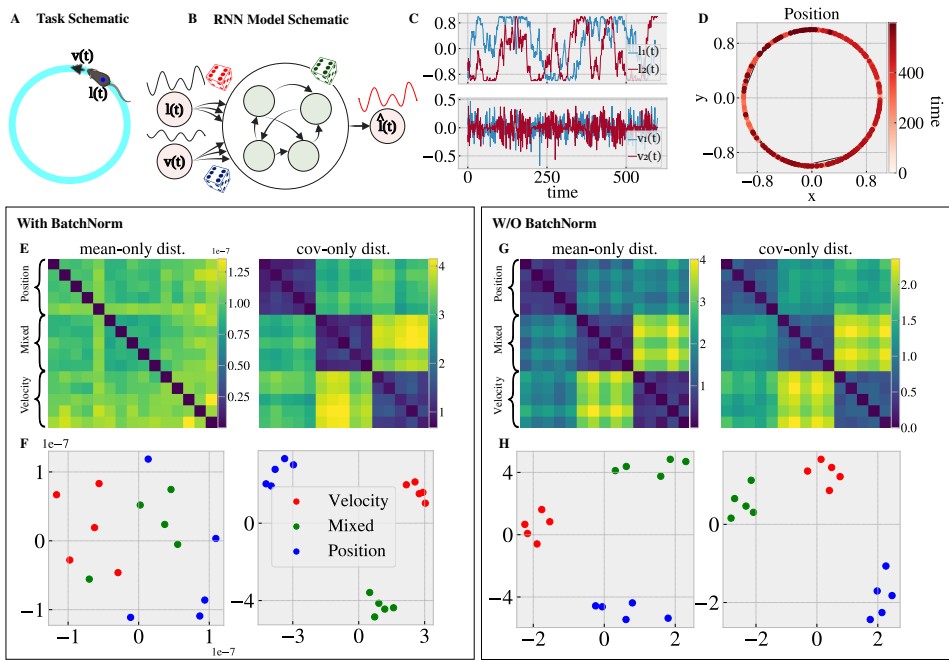

**Figure 2:** SSDs capture differences between recurrent dynamics in high-dimensional nonlinear RNNs without trial structure. *(A)* An animal runs around a circular track. The 2D position and velocity of the animal at time $t$ are $\boldsymbol{l}(t)$ and $\boldsymbol{v}(t)$. *(B)* We trained 3 classes of RNNs with noisy hidden units to estimate the position of the animal at time $t$ with varying levels of velocity and position noise. *(CD)* Sample trajectories of the true position $\boldsymbol{l}(t)$ and velocity $\boldsymbol{v}(t)$. *(E)* Distance matrices show that SSD with $\alpha = 0$ captures the block structure corresponding to 3 types of trained networks (with BatchNorm) while SSD with $\alpha = 2$ fails. *(F)* Multidimensional scaling (MDS) plots of the distance matrices show that differences in the covariance structures cluster the 3 networks, while the distances of the mean responses do not. *(GH)* When BatchNorm is not applied, both the SSD with $\alpha = 0$ and SSD with $\alpha = 2$ capture differences between the networks.

can we disentangle recurrent neural dynamics between networks in such setups? As an example, suppose we have recordings from place cells in two mice while they are freely exploring an environment. Since the two mice will traverse unique trajectories, comparing time-aligned representations may not be a relevant measure for disentangling recurrent interactions. Instead, a more natural approach is to compare representations conditioned on position.

We trained 3 classes of RNNs with 64 hidden units to estimate the position $\boldsymbol{l}(t)$ of an animal on a circular track (Figure 2A). Each RNN received noise corrupted inputs $(\boldsymbol{l}(t) + \epsilon_l \boldsymbol{w}_l(t), \boldsymbol{v}(t) + \epsilon_v \boldsymbol{w}_v(t))$, where $\boldsymbol{v}(t)$ is the velocity of the animal and $\boldsymbol{w}_l(t)$, $\boldsymbol{w}_v(t)$ are standard Gaussian random vectors (Figure 2B). The first (second, third) class of RNNs was trained with $\epsilon_l = 0$ ($\epsilon_l = 0.5$, $\epsilon_l = 1$) and $\epsilon_v = 1$ ($\epsilon_v = 0.5$, $\epsilon_v = 0$). Intuitively, since the first class receives noiseless position inputs, the RNN should simply output its position inputs (position networks). On the other hand, the third class has access to noiseless velocity inputs, so the RNN should integrate these inputs to estimate the position (velocity networks). Finally, we expect the second class learns an intermediate strategy (mixed networks). Therefore, we expect that the 3 classes of RNNs will have distinct recurrent interactions that can be distinguished by comparing their conditional responses.

For each class, we trained the model with and without BatchNorm in their hidden-to-output layer. Since BatchNorm normalizes hidden state activations in each batch, we expect the networks trained with BatchNorm to have near zero trial averages at each time point. Therefore using BatchNorm is akin to adversarial examples in Fig. 1 where the trial averages are forced to be similar across models. Hence we expect SSD with $\alpha = 2$ to fail distinguishing between 3 classes of RNNs.

To confirm these intuitions, we computed SSDs between the networks using eq. (1) by treating the true position as "landmarks." Specifically, we partitioned the ground truth position $\boldsymbol{l}(t)$ into $M$ bins and treated each bin as a landmark. We then computed the mean and covariance of a networks response conditioned on each landmark and compared their SSDs. Consistent with our intuition,

we observed that for BatchNorm RNNs, the SSD with $\alpha = 2$ fails to distinguish the position, velocity, and mixed networks. While the SSD with $\alpha = 0$ successfully distinguishes the 3 classes of RNNs. This is expressed by the clustering of network types in the MDS embedding of the RNNs in Figure 2E and by the block structure of the SSD matrices in Figure 2F.

Conversely, we observed that for RNNs trained without BatchNorm, SSD with both $\alpha = 0$ and $\alpha = 2$ distinguishes between different classes of RNNs (Figure 2G). This suggests that both trial averages and noise correlations contain information about the recurrent strategies used by the networks. Details of the RNN training are provided in Appendix A.2.

### 2.4 WHY CONDITIONAL COVARIANCE CAN DISTINGUISH RECURRENT DYNAMICS WITHOUT TRIAL STRUCTURE

Figure 2 shows that we can leverage conditional covariance structure to distinguish recurrent interactions in a network when there isn't explicit trial structure. To better understand this, let $z(t)$ be a stochastic process that encodes the behaviorally relevant variables (e.g., position or velocity). Here we assume $z(t)$ is scalar valued, but the set-up readily extends to the case of a vector-valued process. Consider the following stochastic differential equation modulated by $z(t)$:

$$d\boldsymbol{x}(t) = f(\boldsymbol{x}(t))dt + u(z(t))dt + g(\boldsymbol{x}(t))d\boldsymbol{w}(t),$$

where $u : \mathbb{R} \mapsto \mathbb{R}^N$ is the function that maps the behavioral variable to the dynamics and all other variables are as in eq. (2). The covariance $\boldsymbol{P}(t)$ evolves according to the differential equation:

$$\frac{d\boldsymbol{P}}{dt} = \mathbb{E}\left[(f(\boldsymbol{x}) + u(z))(\boldsymbol{x} - \boldsymbol{m})^\mathsf{T}\right] + \mathbb{E}\left[(\boldsymbol{x} - \boldsymbol{m})(f(\boldsymbol{x}) + u(z))^\mathsf{T}\right] + \frac{1}{2}\mathbb{E}\left[g(\boldsymbol{x})g(\boldsymbol{x})^\mathsf{T}\right],$$

which now includes an explicit dependence on the time-varying input to the system $u(z(t))$ because $z(t)$ itself is a stochastic process that is not independent of $\boldsymbol{x}(t)$. One way to obtain an equation that does not include the input to the system is by conditioning on the process $z = \{z(t) \geq 0\}$:

$$\boldsymbol{m}(t, z) := \mathbb{E}[\boldsymbol{x}(t)|z], \qquad \text{and} \qquad \boldsymbol{P}(t, z) := \mathbb{E}\left[(\boldsymbol{x}(t) - \boldsymbol{m}(t, z))(\boldsymbol{x}(t) - \boldsymbol{m}(t, z))^\mathsf{T}|z\right].$$

In this case, the explicit dependence on the input $u(z(t))$ drops out of the evolution equation:

$$\frac{d\boldsymbol{P}(z)}{dt} = \mathbb{E}[f(\boldsymbol{x})(\boldsymbol{x} - \boldsymbol{m}(z))^\mathsf{T}|z] + \mathbb{E}[(\boldsymbol{x} - \boldsymbol{m}(z))f(\boldsymbol{x})^\mathsf{T}|z] + \frac{1}{2}\mathbb{E}[g(\boldsymbol{x})g(\boldsymbol{x})^\mathsf{T}|z],$$

which suggests that conditioning on $z$ is useful for isolating the recurrent contribution to the dynamics. In the above, we have conditioned on the *entire* process $z$, so comparing two networks' responses requires that the behavioral processes have similar trajectories. However, if we further assume that the pair $(\boldsymbol{x}, z)$ is a *stationary* process—i.e., the joint distribution of $(\boldsymbol{x}(t), z(t))$ does not depend on time $t$—then the conditional mean and covariance, defined by $\boldsymbol{m}(z) := \mathbb{E}[\boldsymbol{x}|z]$ and $\boldsymbol{P}(z) := \mathbb{E}\left[(\boldsymbol{x} - \boldsymbol{m}(z))(\boldsymbol{x} - \boldsymbol{m}(z))^\mathsf{T}|z\right]$, do not depend on time $t$ and can be used to compare representations from networks with different trajectories as in the previous section.

## 3 DISCUSSION

We have shown that SSDs can be used to disentangle noisy dynamic systems with different recurrent interactions. As shown by Ostrow et al. (2023), we find that the mean trajectories of networks with different recurrent interactions can be quite similar due to external inputs to the system. However, we find that comparing stochastic fluctuations about the mean trajectories can distinguish between systems with differing recurrent interactions, as demonstrated by (Galgali et al., 2023). By using the full parametric form of SSDs (with $\alpha$ interpolating between 0 and 2), we can obtain a richer notion of the similarities and differences, quantified in terms of metrics, in recurrent interactions between networks with noisy dynamics.

There are many important directions for future investigation. For example, the SSDs used in this work only capture first and second-order moments, which is optimal when comparing Gaussian processes; however, they are not guaranteed to capture differences between non-Gaussian processes. Another important avenue to explore is the relationship between SSDs measures of (dis)similarity discussed here and DSA measures introduced by Ostrow et al. (2023). SSDs are metrics designed to capture similarities in *geometric* structure, whereas DSA has been shown to capture similarities in *topological* structure. Teasing apart these details is critical to enriching our understanding of the dynamics at play in neural systems.

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

## A  EXPERIMENTAL DETAILS

### A.1  THREE NETWORK DYNAMICS

We modeled three network dynamics—a bistable switch, a line attractor, and a point attractor—that were analyzed in (Galgali et al., 2023) and then in (Ostrow et al., 2023). To simulate the trajectories, we used the code from the Jupyter notebook developed by Ostrow et al. (2023), which can be found within the following Github repository: `https://github.com/mitchellostrow/DSA`. To compute the SSDs, we used the code from the following Github repository: `https://github.com/ahwillia/netrep`. The dynamics and simulation details are described below.

#### A.1.1  NONLINEAR BISTABLE SWITCH

The bistable switch is described by the 2-dimensional system of nonlinear stochastic differential equations with additive (degenerate) noise

$$dx(t) = [f(x(t)) + 1\nu_i]dt + \sigma dw(t),$$

where $f(x_1, x_2) := (ax_1^3 + bx_1, cx_2)$, $1 = (1, 1)$ is the vector of ones, $\nu_{\text{cond}}$ is a conditioned dependent constant input-drive that is applied to both coordinates, $w(t)$ is a 2-dimensional standard Brownian motion and $a, b, c, \sigma$ are constants. The parameters $a, b, c$ are randomly sampled at the outset as follows: $a \sim \text{Unif}(-5, -3)$, $b \sim \text{Unif}(4, 7)$, $c \sim \text{Unif}(-4, -2)$. Depending on the condition, $\nu_{\text{cond}}$ is set to $-0.1$ or $0.1$ and drives the system to one of the stable fixed points. The noise coefficient $\sigma$ is 5. The system is simulated using the Euler-Maruyama method:

$$x_{n+1} = x_n + f(x_n)\Delta t + 1\nu_i\Delta t + \sigma\Delta w_n$$

with step size $\Delta t = 0.01$ and $\Delta w_n$ are i.i.d. with $\Delta w_{n,i} \sim \mathcal{N}(0, \Delta t)$ for $i = 1, 2$.

#### A.1.2  LINEAR LINE ATTRACTOR AND POINT ATTRACTOR

The line attractor and point attractor are 2-dimensional linear stochastic differential equations of the form

$$dx(t) = [Ax(t) + u(t)]dt + \sigma dw(t),$$

where $w$ is a 2-dimensional standard Brownian motion and $\sigma$ is as in the previous example. For the linear attractor, the recurrent dynamics matrix is $A = V\Lambda V^{-1}$, where $\Lambda = \text{diag}(-1, 0)$ and

$$V = \begin{bmatrix} 1 & 1 \\ 1 & 0 \end{bmatrix}.$$

For the point attractor, the recurrent dynamics matrix is $A = \text{diag}(-0.5, -1)$. As in previous example, each linear system is simulated using the Euler-Maruyama method with In each case, the input drive $u(t)$ is adversarially chosen to minimize the difference between the mean trajectory of the linear system with the mean trajectory of the nonlinear bistable switch. In particular, let $\bar{x}_n^{\text{BS}}$ denote a conditional mean trajectory of the bistable switch (i.e., conditioned on $u = 1$ or $u = -1$) and let $\bar{x}_n$ denote the corresponding condition mean trajectory of the linear system. Then the input drive for the linear system $u_n$ and the mean trajectory for the linear system can be recursively defined by $\bar{x}_0 = \bar{x}_0^{\text{BS}}$ and

$$u_n = \frac{\bar{x}_{n+1}^{\text{BS}} - \bar{x}_n}{\Delta t} - A\bar{x}_n, \qquad \bar{x}_{n+1} = \bar{x}_n + [A\bar{x}_n + u_n]\Delta t.$$

### A.2  RNN

The synthetic dataset was simulated to model the behavior of a mouse running around a circular track. The task is motivated by neuroscience findings on the existence of a ring attractor in the

representations of entorhinal neurons and RNN models trained to track the position of animals (Low et al., 2021). To generate the position and velocity of the fictitious animal in time, we first sampled a parameter $\theta_t$ from a Brownian motion for $600$ time steps with a drift of $0.2$. The 2D position at each time step was calculated according to $\boldsymbol{l}_t = (\cos \theta_t, \sin \theta_t)$. The 2D velocity was then computed as the discrete differential of the position $\boldsymbol{v}_t = \boldsymbol{l}_t - \boldsymbol{l}_{t-1}$. Each RNN has 4 inputs and 2 outputs and is tasked to estimate the position of the animal denoted by $\hat{\boldsymbol{l}}_t$. Therefore we minimize the following loss function $\mathcal{L}(\boldsymbol{\phi}) = \sum_{t,i} \|\boldsymbol{l}_{t,i} - \hat{\boldsymbol{l}}_{t,i}\|_2^2$ where $t$ corresponds to the time point $t$ and $i$ denotes the trial $i$. Each network has $64$ hidden units and follows the classical rate network dynamics:

$$\boldsymbol{h}_{t+1} = \sigma(\boldsymbol{W}_h \boldsymbol{h}_t + \boldsymbol{W}_i (\boldsymbol{l}_t + \boldsymbol{\epsilon}_t^l, \boldsymbol{v}_t + \boldsymbol{\epsilon}_t^v)^T + \boldsymbol{\epsilon}_t^h), \quad \hat{\boldsymbol{l}}_t = \boldsymbol{W}_o \boldsymbol{h}_t, \quad \boldsymbol{\phi} = \{\boldsymbol{W}_h, \boldsymbol{W}_i, \boldsymbol{W}_o\}$$

where $\boldsymbol{\epsilon}_t^l, \boldsymbol{\epsilon}_t^v, \boldsymbol{\epsilon}_t^h$ are i.i.d. noise terms added to the position input, velocity input, and hidden units respectively. The standard deviation of these noise terms determines how much the network can rely on each of the two input terms. For 5 velocity networks, we set the noise standard deviation of the position and velocity inputs to 1 and 0 respectively. We do the opposite for 5 position networks. For 5 mixed networks with injected noise with standard deviation $0.5$ to both position and velocity inputs. The hidden noise standard deviation is always set to $0.1$.

For each class of RNNs (position, velocity, and mixed) we trained two separate models, with and without BatchNorm in their hidden-to-output layers.

$$\boldsymbol{h}_t \leftarrow \frac{\boldsymbol{h}_t - \mathbb{E}[\boldsymbol{h}_t^{\text{batch}}]}{\sqrt{\text{Var}(\boldsymbol{h}_t^{\text{batch}})}}, \quad \hat{\boldsymbol{l}}_t = \boldsymbol{W}_o \boldsymbol{h}_t$$

Since BatchNorm normalizes hidden state activations to have zero mean, it intuitively mimics an adversarial training strategy for limiting the information of the trial averages.

To train the networks, we generated batches of size $64$ and ran $1000$ iterations of backprop through time using `SGD` optimizer with a learning rate of $0.05$. All the experiments were done using `Pytorch` on CPU and every RNN takes about 3 minutes to train.

