# OpenReview forum: "Disentangling Recurrent Neural Dynamics with Stochastic Representational Geometry"
_ICLR.cc/2024/Workshop/Re-Align — ICLR 2024 Workshop Re-Align ContributedTalk_

### Official Review · Reviewer_F9PZ · 2024-02-23
**SSDs as an improved measure of neural dynamic (dis)similarity**

**Rating:** 3
**Fit:** 3
**Confidence:** 2

**Workshop Review:**

Summary:
The authors demonstrate that the Stochastic Shape Distance, a measure of distance between two gaussian distributions is useful for disentangling neural dynamics which come from fundamentally different recurrent systems, but otherwise may be considered identical under other distance metrics. Specifically, first they show (both empirically and analytically) that by considering across-trial covariance they can disentangle systems with the same trial averaged trajectories. Secondly, they show how even without trial structure, the conditional covariance can be leveraged to a similar effect.

Strengths:
- (Clarity) The paper is well written, the idea is thoroughly explored and well situated within the literature.
- (Correctness) The experiments and analysis appear to justify the papers main claims sufficiently.
- (Novelty) The proposed application of the very recently introduced method is novel to the best of my knowledge.
- (Relevance) There is certainly relevance to designing accurate measures of dynamical similarity, and this is a good step in that direction.

Weaknesses:
- The experimental details are quite minimal making the paper not very self-contained, with frequent references to external works for clarity. For an outstanding paper, it would be beneficial for the authors to include all these details in the appendix.
- The method does appear helpful in the situations the authors constructed, however it is not clear to what extent this method will remain beneficial in natural settings or if this if the presented examples are pathological. For example, does the SSD really capture meaningful variation between trial-to-trail differences in natural data, or are higher-order moments necessary as the authors allude to?

Minor:
- (Typo) Middle of Page 5, "One way to obtain an equation that does not include the input to the system is by conditioned on ..." --> "... is by conditioning on..."

**Reason For Not Giving Higher Score:**

N/A

**Reason For Not Giving Lower Score:**

The paper's topic is very timely and clearly in-line with the workshop goals. It includes analytical and empirical justification, and a new viewpoint on dynamic similarity. There are no flaws with the paper to my view other than the potential that more could be done to improve the method.

**Reviewer Domain:**

machine learning

---

### Official Review · Reviewer_EYpj · 2024-02-24
**The paper presents novel applications of stochastic shape distances (SSDs; Duong et al., 2023)  to disentangle recurrent neural dynamics**

**Rating:** 3
**Fit:** 3
**Confidence:** 2

**Workshop Review:**

**Overall summary:**

The paper is very well written and its subject is relevant to the community of this workshop. The experiments are simple yet effective in supporting the paper's claims, which are also supported by theoretical arguments that, to the best of my understanding, seem correct. A suggestion is to add a bit more details for the experiments in section 2.1 (or at least add them in the appendix) so the reader doesn't have to refer to past works for something that is of key importance in the present paper.

**Typos:**

- Possible error in this sentence? “The first class of RNNs was trained on noiseless position inputs and noisy velocity inputs, whereas the second class of RNNs was trained on noiseless velocity inputs and noisy position inputs (Figure 2B)”

**Reason For Not Giving Higher Score:**

N/A

**Reason For Not Giving Lower Score:**

The paper is very well written, the topic is of interest to the community, and the claims are supported both by empirical results and by theoretical arguments. It's a high quality workshop paper.

**Reviewer Domain:**

neuroscience

---

### Official Review · Reviewer_1qzR · 2024-02-24
**Review of "Disentangling Recurrent Neural Dynamics with Stochastic Representational Geometry"**

**Rating:** 3
**Fit:** 3
**Confidence:** 2

**Workshop Review:**

**Summary**

The paper shows that recent advances in shape metrics, specifically stochastic shape distances (SSDs), can distinguish different underlying dynamics even when the trial-averaged trajectories are similar. It analyses the interplay between stochasticity and recurrent dynamics and how the former can aid in comparing the latter. The authors provide simple experiments and interesting theoretical analyses to demonstrate this.

**Strengths**
* The paper is well-written and provides interesting insight on how SSDs, despite not explicitly modelling dynamics, can still be used to distinguish them.
* The numerical experiments are intuitive and comprehensive.

**Questions and suggestions**
* It would be nice if the authors could comment on whether SSDs could be used to distinguish different learning rules, i.e., dynamics of learning itself, similar to Ostrow et al. (2023).
* Perhaps showing some of the failure modes of these SSDs in distinguishing different networks would aid readers in understanding where best to use or not use these metrics.
* As the authors state, it would be interesting to explore the connections between DSA (Ostrow et al., 2023), which performs a modified Procrustes alignment of DMD matrices, and the SSDs which align network responses based on their moments.

Overall, I believe that this work would be of great interest to the workshop's participants, and I recommend that the paper be accepted.

**Reason For Not Giving Higher Score:**

N/A

**Reason For Not Giving Lower Score:**

The submission demonstrates how recent advances in shape metrics can distinguish different dynamics. This builds upon a line of work that is highly relevant to the workshop, with several impactful future directions. Furthermore, the submission is well-written and provides a good mix of theoretical and empirical analyses.

**Reviewer Domain:**

machine learning

---

### Decision · Program_Chairs · 2024-03-02

Accept (Contributed Talk)